# scRNA+TCR-seq reveals the proportion and characteristics of dual TCR Treg cells in mouse lymphoid and non-lymphoid tissues

Yuanyuan Xu, Qi Peng, Xiaoping Lu, Long Ma, Jun Li, Xinsheng Yao*

Department of Immunology, Zunyi Medical University; Guizhou Provincial Research Center for Applied Immunomolecular Engineering; Guizhou Provincial Innovation Base for Graduate Education in Immunology; Guizhou Provincial Key Discipline of Immunology, Zunyi, China

## eLife Assessment

This study reanalyzed previously published scRNA-seq and TCR-seq data to examine the proportion and characteristics of dual-TCR-expressing Treg cells in mice, presenting some **useful** insights into TCR diversity and immune regulation. However, the evidence is **incomplete**, particularly with respect to data interpretation, statistical rigor, and the functionality of dual -TCR Treg cells. The study is potentially of interest to immunologists studying T-cell biology.

*For correspondence:
immunology@126.com

Competing interest: The authors declare that no competing interests exist.

**Abstract** The rearrangement of TCR germline V(D)J genes during T cell development, including allelic exclusion and tolerance selection, ensures the clonal selection theory, which states that 'a lymphocyte expresses only one type of antigen receptor'. This forms the basis for T cell-specific responses. However, the existence of 'dual TCR T cells' has consistently been supported by specific experimental evidence. Detailed reports on the origin, proportion, tissue distribution, and CDR3 characteristics of 'dual TCR Treg cells' are currently lacking. In this study, we utilized scRNA+T-CR-seq technology to achieve in-depth analysis of single and dual TCR T pairings, along with their mRNA expressions, from over 5000 T cells in each sample. Through comparative studies with shared databases, we provided a detailed analysis of the proportions and characteristics of dual TCR Tregs in mouse lymphoid and non-lymphoid tissues (such as inguinal lymph node, mesenteric lymph node, blood, and skin). Our findings revealed a high proportion of dual TCR Tregs across various mouse tissues, with their TCR pairing patterns, V(D)J usage, and mRNA expression showing both homogeneity and certain differences compared to single TCR Tregs, as well as heterogeneity across different tissue sites. This research provides new insights and technical approaches for studying the origins, characteristics, effects, and mechanisms of Treg cells in different tissue locations.

## Introduction

The discovery of CD4+CD25+ Treg cells in 1995 has since been linked to the occurrence and development of various human diseases. In addition to their strong regulatory role in self-reactivity and excessive inflammation, Treg cells play an important homeostatic role in the maintenance of tissue homeostasis, muscle damage repair, hair follicle regeneration, fat metabolism, among other complex cellular activities (*Sakaguchi et al., 2020*; *Shevyrev and Tereshchenko, 2019*). The pleiotropic suppressive functions of Treg cells imply distinctive characteristics in TCR diversity, antigen specificity,

and memory formation. However, the compositional characteristics of Treg cell TCR, their dependency on antigen epitopes, activation, and maintenance mechanisms, remain largely unelucidated. Some studies have garnered widespread attention, including V(D)J recombination of nTreg cells TCRs, self-tolerance selection, and migratory settlement with or without lineage characterization; differences in antigen induction and activation mechanisms for iTreg versus conventional helper and effector T cells; diversity and plasticity of Treg cells in lymphoid and non-lymphoid tissues, such as TCR CDR3, and their correspondence to regulatory functions.

Since their initial report in 1988 (*Malissen et al., 1988*), dual TCR-expressing lymphocytes have been widely supported by experimental evidence and play important roles in physiological and pathological processes such as autoimmune tolerance, transplant rejection reactions, and T cell tumors (*Padovan et al., 1993*; *Hinz et al., 2001*; *He et al., 2002*; *Schuldt and Binstadt, 2019*; *Zhu et al., 2023*; *Xu et al., 2024*; *Schuldt et al., 2017*). *Tuovinen et al., 2006* utilized flow cytometry to discover that the frequency of dual TCR expression in human CD4+CD25+ Treg cells is higher than in other T cell subgroups, and that dual TCR Treg cells can effectively transmit signals, suggesting that dual TCR expression may promote the development and function of Treg cells. However, this study primarily focused on the dual TCR α chain, leading to certain limitations in its conclusions. Overall, due to technical constraints, the findings are limited in comparing the proportion, characteristics, origin, and mechanisms of single and dual TCR T cells in a single sample of more than a thousand T cells, posing challenges for immunologists. However, scRNA-seq combined with scTCR-seq can mark over 5000 T cells from a single research sample at once, obtaining the pairing types of each T cell's TCR beta chain and alpha chain, CDR3 sequence composition characteristics, and complete mRNA expression data. This allows for the accurate analysis of TCR CDR3 characteristics in the development, differentiation, maturation, and response/tolerance processes of each T cell, as well as the corresponding expression of regulatory and effector molecules. This provides unprecedented opportunities for in-depth analysis of dual receptor T cells (*Zhu et al., 2023*; *Xu et al., 2024*; *Papalexi and Satija, 2018*; *Park et al., 2020*).

First, in the Treg study conducted by *Burton et al., 2024*, which involved analyzing single-cell RNA sequencing and T cell receptor (scRNA+TCRseq) data of Foxp3+ Treg cells sorted from mouse blood, kidney, liver, LPL, and pancreas under physiological conditions, we conducted a detailed comparative analysis of the proportions of single and dual TCR Treg; the usage, diversity, clonality, and overlap of CDR3 VJ; and the homogeneity and heterogeneity of characteristic marker mRNA expression. Additionally, to further explore the differences in dual TCR Treg cells between lymphoid and non-lymphoid tissues in mice, we selected the scRNA+TCR-seq data shared by *Nedwed et al., 2023*, which includes sequencing results of CD4+CD25+ Treg cells sorted from mouse spleen, inguinal lymph nodes (iLN), and mesenteric lymph nodes (mLN) under physiological conditions, as well as CD3+T cells (including Treg and non-Treg) sorted from skin tissue, and performed comparative analysis again.

## Results

### A high proportion of dual TCR Treg cells in mouse lymphoid and non-lymphoid tissues

The proportion of dual TCR Treg cells in mouse lymphoid tissues: spleen = 21.4%, iLN = 21.4%, and mLN = 21.0% (Figure 3D); the proportion of dual TCR Treg cells in mouse non-lymphoid tissues: blood = 11.8%, kidney = 12.2%, liver = 13.6%, LPL = 11.0%, and pancreas = 13.3%, the proportion of dual TCR T cells in mouse skin tissue: 18.6% (*Figure 1D*)(*Figure 2D*). Subsequently, a detailed analysis of the pairing of dual TCR T cells in the two sets of data was conducted, revealing that the TCR pairing types of dual TCR Treg include A+B1+B2, B+A1+A2 (the highest proportion), and A1+A2+B1+B2 (the lowest proportion). Significant differences in TCR pairing types exist between tissues, with A1+A2+B1+B2 most highly expressed in skin (common in non-Treg cells of skin) and kidney, and B+A1+A2 most highly expressed in spleen and liver compared to other tissues (*Figure 1D*; *Figure 2D*).

### Homogeneity and heterogeneity of TCR composition in mouse dual TCR Treg cells VJ usage (*Figure 1E, F* and *Figure 2E,F*)

The overall VJ family distribution pattern of single and dual TCR Treg between different tissues is basically consistent. Compared with single TCR Treg, dual TCR Treg displays specific preferences in V

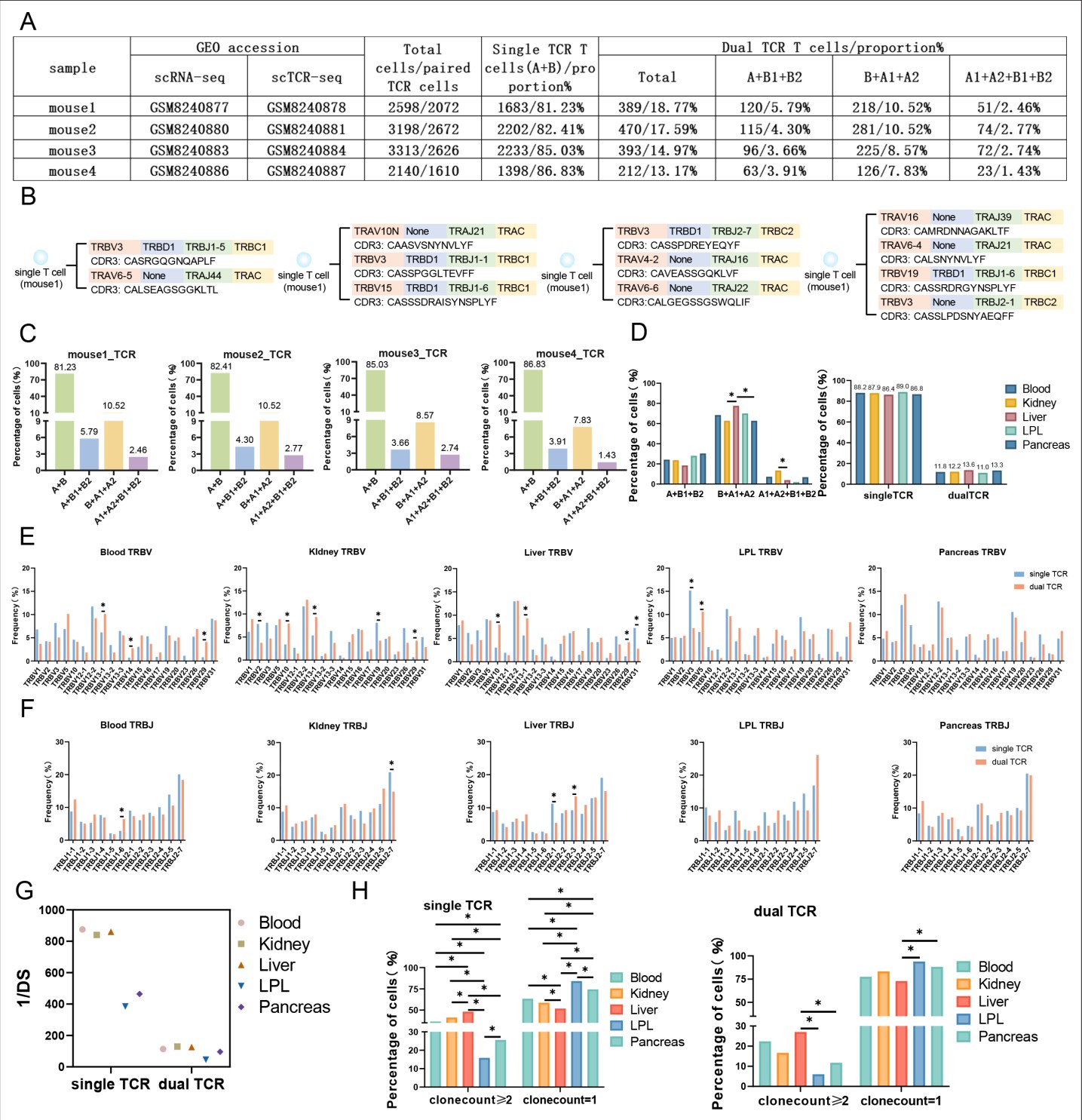

**Figure 1.** Characterizes single Treg TCR and dual TCR Treg in the blood, kidney, liver, LPL, and pancreas of mice. (**A**) Mouse source names, GEO accession number, total number of paired TCR Treg cells, and number and proportion of TCR paired types. (**B**) Four different TCR pairing types, VDJ gene family names and CDR3 AA sequences in single T cells. (**C**) Proportion of TCR pairing types per mouse. (**D**) Statistical analysis of dual TCR Treg cells and the relative proportion of single/dual TCR Treg cells in three TCR pairing types. (**E**) Comparative analysis of single/dual TCR Treg cell *V* gene usage variation. (**F**) Comparative analysis of single/dual TCR Treg cell *J* gene usage variation. (**G**) Differential analysis of single/dual TCR Treg cell diversity among tissues. (**H**) Differential analysis of single/dual TCR Treg cell clone expansion between tissues.

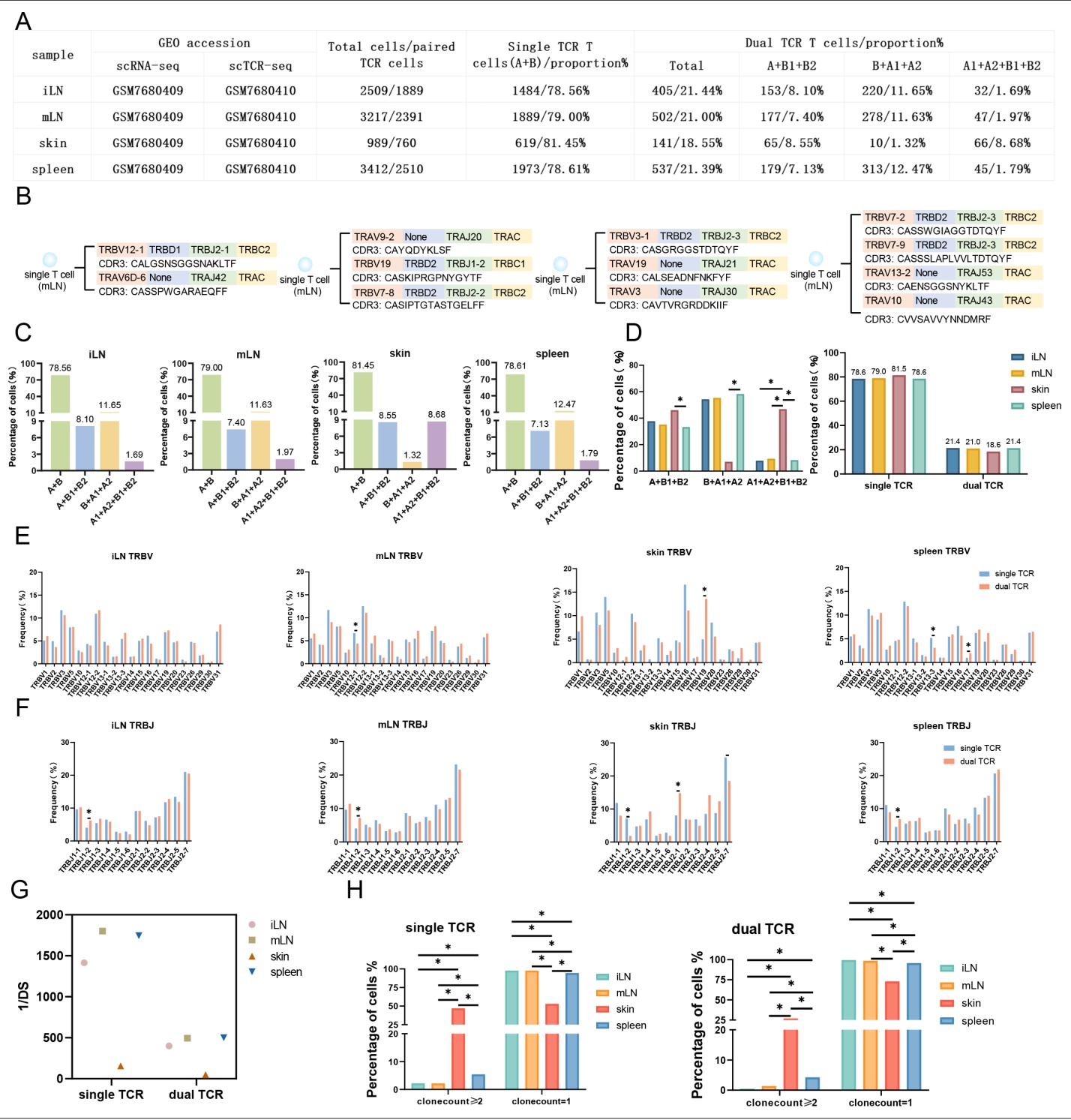

**Figure 2.** Characterizes single Treg TCR and dual TCR Treg in the inguinal lymph node (iLN), mesenteric lymph node (mLN), skin, spleen of mice. (**A**) Tissue source names, GEO accession number, total number of paired TCR T cells, and number and proportion of TCR paired types. (**B**) Four different TCR pairing types, VDJ gene family names, and CDR3 AA sequences in single T cells. (**C**) Proportion of TCR pairing types per tissue. (**D**) Statistical analysis of dual TCR T cells and the relative proportion of single/dual TCR T cells in three TCR pairing types. (**E**) Comparative analysis of single/dual TCR T cell V gene usage variation. (**F**) Comparative analysis of single/dual TCR T cell J gene usage variation. (**G**) Differential analysis of single/dual TCR T cell diversity among tissues. (**H**) Differential analysis of single/dual TCR T cell clone expansion between tissues.

and J usage, mainly including *Trbv10* and *Trbv13-1* in the kidney and liver, *Trbv5* in LPL, *Trbv19* in the skin; *Trbj2-3* in the liver, *Trbj2-1* in the skin, etc. showing a significant advantage in usage; *Trbv13-1* in peripheral blood, kidney, and liver shows consistent advantage in usage; *Trbj1-2* in the spleen, iLN, and mLN shows consistent advantage in usage, etc.

### Diversity of CDR3 (*Figure 1G* and *Figure 2G*)
The 1/DS of single TCR Treg CDR3 is significantly higher than that of dual TCR Treg; the CDR3 diversity of dual TCR Treg in lymphoid tissues (spleen, iLN, and mLN) is higher than that in non-lymphoid tissues; the CDR3 diversity of dual TCR Treg in skin, LPL, and pancreas is very low.

### Clonality of CDR3 (*Figure 1H* and *Figure 2H*)
The proportion of single TCR Treg with clonal proliferation greater than or equal to 2 is higher than that of dual TCR Treg; the proportion of dual TCR Treg with clonal proliferation greater than or equal to 2 in the liver and skin is significantly higher than that in other tissues.

### Overlap of CDR3 (*Figure 3A,B* and *Figure 4A ,B*)
A certain proportion of overlap exists between TRB CDR3 AA and TRA CDR3 AA in different tissues, with single TCR Treg significantly higher than dual TCR Treg, and a lower proportion of TRB CDR3 AA compared to TRA CDR3 AA. This suggests that TRA allelic inclusive recombination involves more CDR3 AA types in TCR pairing of Treg cells. Meanwhile, it was found that single and dual TCR Treg in different tissues can also share common CDR3 AA. This suggests that both single and dual TCR Treg can respond to specific antigen epitopes.

All three tissues, the liver, kidney, and pancreas, show a significantly high proportion of dual TCR Treg CDR3 AA overlap with peripheral blood, suggesting their origin from the influx of peripheral blood (it should be noted that the high overlap phenomenon of the kidneys, liver, pancreas, and blood may not completely eliminate the technical possibility of local Treg migration within the blood), iLN, mLN, and skin all show a significantly high proportion of dual TCR Treg CDR3 AA overlap with the spleen, suggesting their origin from the influx of the spleen; Intriguingly, there is also a high proportion of dual TCR Treg CDR3 AA overlap between the liver and kidney, kidney and pancreas, and iLN and mLN, suggesting that dual TCR Treg cells in these tissue sites may negatively regulate a shared set of antigens. Whereas the mechanism and significance of the high percentage of overlap exhibited between iLN and skin TRA CDR3 AA deserve further analysis.

## Homogeneity of mouse dual TCR Treg cell phenotypes
Comparative analysis of Treg cells with complete scRNA-seq and scTCR-seq results showed that the TCR pairing types and proportion were consistent with the results of direct analysis of scTCR-seq; the proportion of dual TCR Treg in the skin was higher than that of dual TCR non-Treg (*Figure 3C*) (*Figure 4C*); in all tissues, the expression of signature molecules *Tnfrsf9*, *Stat3*, *Tnfrsf4*, *Cd27*, *Icos*, *Il2ra*, *Ikzf2*, *Lrrc32*, *Sh3rf1*, *Cd81*, *Baff*, *Smad2*, and *Foxo3* were identical between dual and single TCR Treg in all tissues, with slightly higher *Foxp3*, *Foxo1, and Ctla4* expression in dual TCR Treg than that in single TCR Treg (*Figure 3D*)(*Figure 4D*).

## Heterogeneity of mouse dual TCR Treg cell phenotypes
Compared with iLN, mLN, and spleen, more characteristic mRNA molecules were highly expressed in dual TCR Tregs in skin (*Figure 4E*), such as *Rora*, *Dusp1*, and *Junb* involved in immune responses and inflammatory diseases; *S100a6* involved in the proliferation, differentiation, and migration of immune cells; and *Hopx* and *Nr4a1* involved in the development of immune cells, suggesting that dual TCR Treg in the skin may be involved in more specific response negative regulations.

Between blood and liver, kidney and pancreas (*Figure 3E*), iLN and mLN and spleen (*Figure 4E*), the mRNA expression of cytokines, cytokine receptors, and transcription factors in dual TCR Treg shows higher homogeneity, suggesting that dual TCR Treg in these tissue sites may play convergent negative immune response regulations.

Differential mRNA expression analysis between dual and single TCR Treg in each tissue (*Figure 3F*) (*Figure 4F*) showed significant high expressions of individual *Trbv* genes in different tissues, consistent with scTCR-seq results, suggesting specific dual TCR Treg lineage origins in different tissues; in

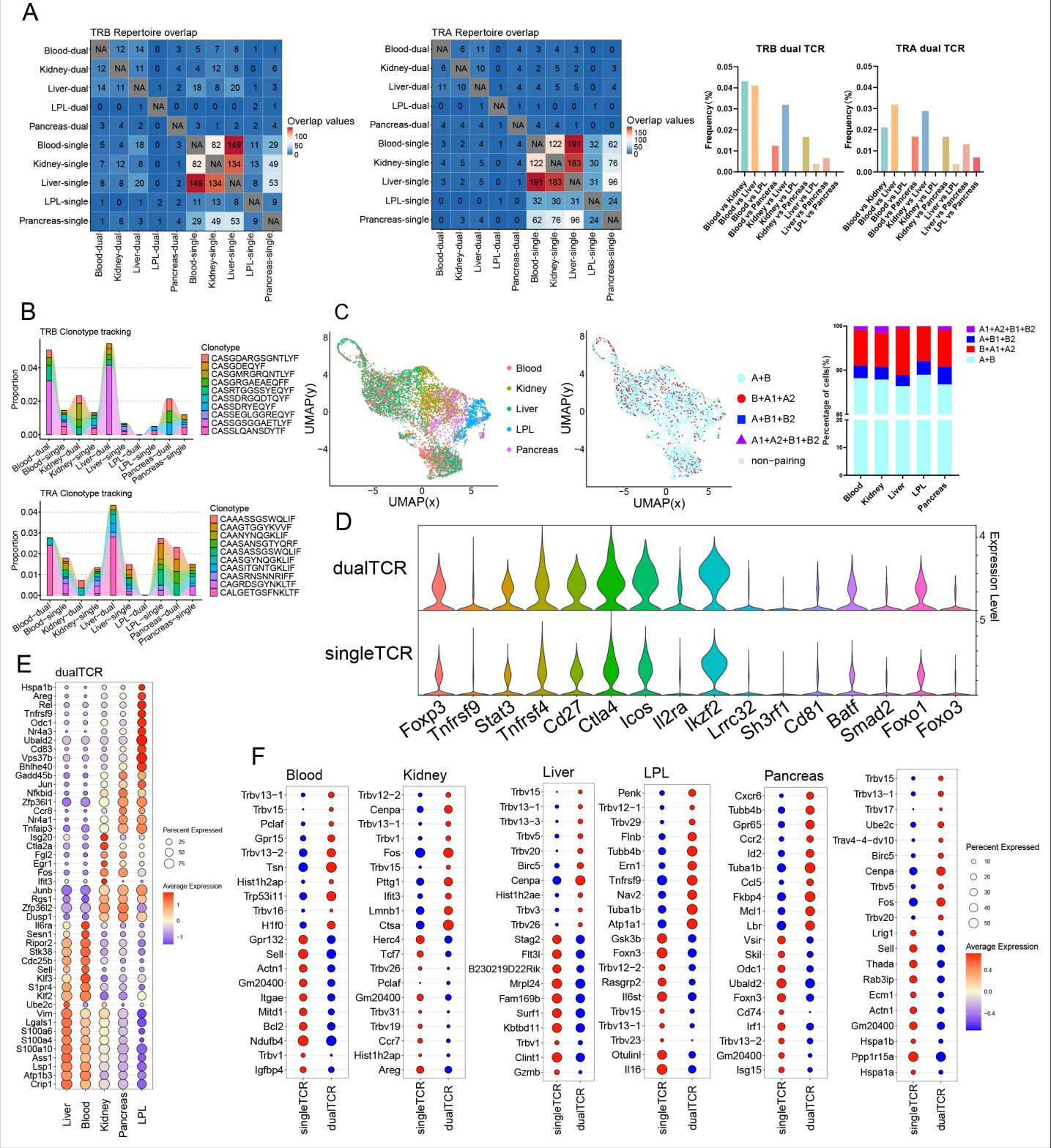

**Figure 3.** Comparative analysis of single Treg TCR and dual TCR Treg CDR3 overlap and gene mRNA expression in mice blood, kidney, liver, LPL, and pancreas. (**A**) Clonal overlap analysis and Jaccard index of TRA/B chain CDR3 region in single/dual TCR Treg cells. (**B**) Tracking results of the first 10 high-frequency overlapping sequences in the CDR3 region of the TRA/B chain of single/dual TCR Treg cells between tissues. (**C**) Treg cell clustering results and proportion of Treg cells of the four TCR pairing types among tissues. (**D**) Characterized mRNA expression in single/dual TCR Treg cells. (**E**) Top 10 mRNA molecules highly expressed by dual TCR Treg cells in each tissue. (**F**) Top 10 mRNA molecules highly expressed by single/dual TCR Treg cells in each tissue and overall.

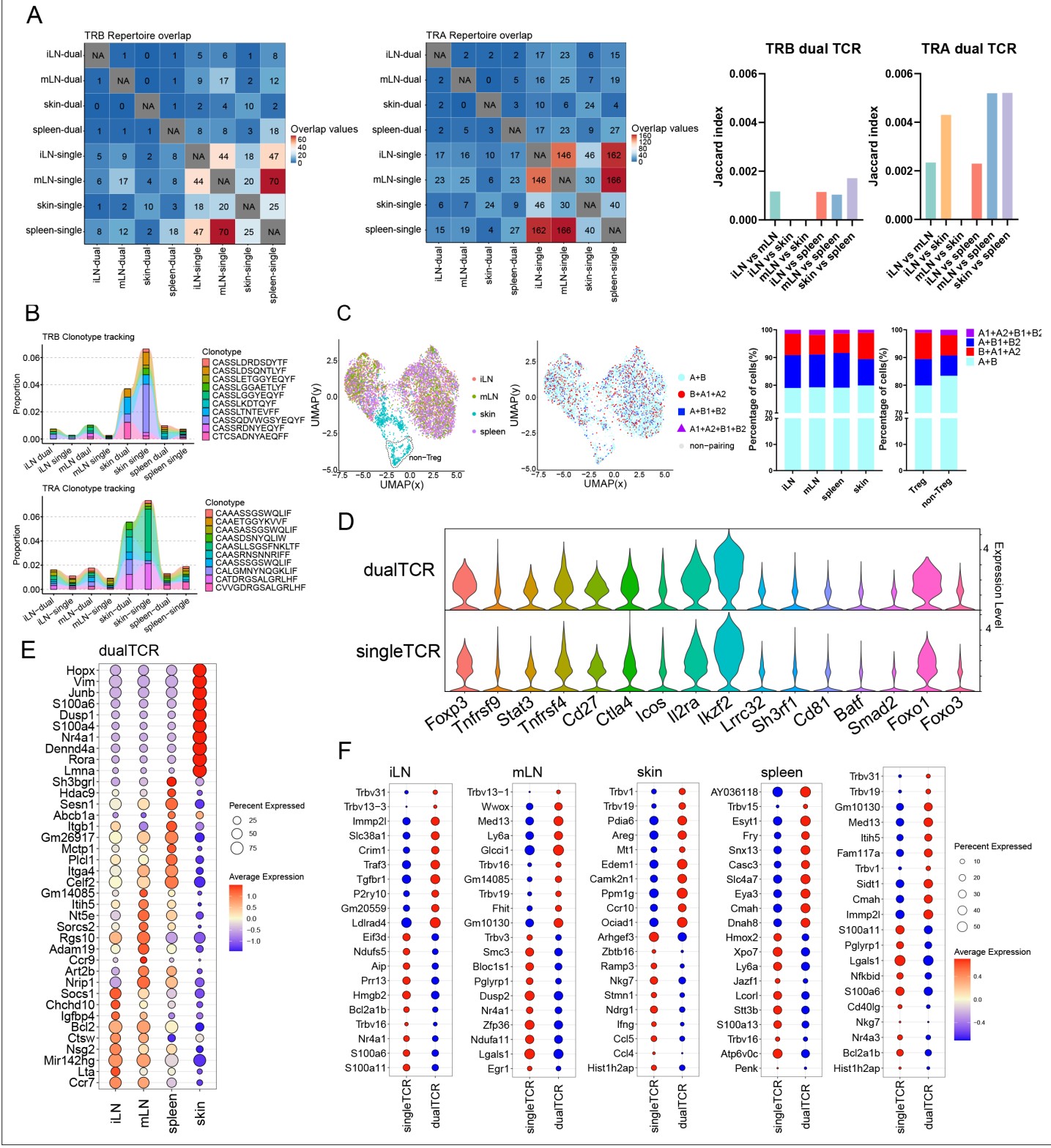

**Figure 4.** Comparative analysis of single Treg TCR and dual TCR Treg CDR3 overlap and gene mRNA expression in mice inguinal lymph node (iLN), mesenteric lymph node (mLN), skin, and spleen. (**A**) Clonal overlap analysis and Jaccard index of TRA/B chain CDR3 region in single/dual TCR T cells. (**B**) Tracking results of the first 10 high-frequency overlapping sequences in the CDR3 region of the TRA/B chain of single/dual TCR T cells between tissues. (**C**) Treg cell clustering results and proportion of Treg cells of the four TCR pairing types among tissues. (**D**) Characterized mRNA expression in single/dual TCR Treg cells. (**E**) Top 10 mRNA molecules highly expressed by dual TCR Treg cells in each tissue. (**F**) Top 10 mRNA molecules highly expressed by single/dual TCR Treg cells in each tissue and overall.

addition to *Trbv* usage, each group's dual TCR Treg had significantly different characteristic molecule expressions from single TCR Treg, such as blood: *H1f0* involved in immune cell proliferation and differentiation and *Hist1h2ap* related to immune cell activation were increased; kidney: *Fos* (p < 0.001) and *Cenpa* (p = 0.003) related to immune cell proliferation and differentiation and *Ifit3* involved in the body's antiviral response were significantly increased; liver: *Cenpa* (p = 0.022) related to immune cell proliferation and *Birc5* involved in the body's anti-infection immune response were increased; LPL: *Flnb* related to cell adhesion and migration regulation and *Atp1a1* related to cell proliferation, differentiation, and apoptosis regulation were increased; pancreas: *Ccl5*, *Ccr2*, and *Cxcr6* related to immune cell migration and *Lbr* (p = 0.009) and *Id2* (p = 0.029) related to immune cell proliferation and differentiation, as well as *Fkbp4* (p = 0.004) involved in immune suppression signal transduction were significantly increased; iLN: *Tgfbr1* involved in maintaining immune tolerance and regulating inflammatory responses and *Traf3* involved in immune cell activation and cytokine production were increased; mLN: *Glcci* (p <0.001) involved in inflammation and immune cell migration was increased; skin: *Areg* (p = 0.008) involved in immune cell proliferation, differentiation, migration, and immune suppression functions and *Ccr10* involved in T cell migration and positioning were increased; spleen: *Casc3* related to T cell activation and proliferation was increased. This suggests that dual and single TCR Treg in each specific tissue site may be involved in different negative immune response regulations.

Consistent high expression of molecular mRNA between different lymphoid tissues, such as the presence of the *Nr4a1* gene related to T cell development, activation, and proliferation in both iLN and mLN tissues; partial sharing of molecular mRNA high expression between different non-lymphoid tissues, such as the expression of *Trbv13-1* and *Trbv5* genes in blood, kidney, and liver tissues, and the common expression of *Trbv1* in skin and kidney, these results may suggest that dual TCR Treg cells in some tissues have specific lineage origins; and the common expression of the *Cenpa* gene in kidney and liver tissues may play a common immune effect; similarly, single TCR Treg also has similar gene expressions, such as the high expression of *Trbv1* in single TCR Treg cells in blood and liver tissues, and the high expression of *Hist1h2ap* related to immune cell activation in skin and kidney.

## Discussion

Currently, the proportion of dual TCR T cells is reported to be extremely variable across physiologic and pathologic states (0.1–30%, etc.) due to differences in assay technology and methodology (*Padovan et al., 1993*; *Hinz et al., 2001*; *He et al., 2002*; *Schuldt and Binstadt, 2019*; *Zhu et al., 2023*; *Xu et al., 2024*; *Schuldt et al., 2017*). This study, in a large number of single Treg cell scRNA+T-CR-seq studies, found that mouse lymphoid and non-lymphoid tissues both have a high proportion of dual TCR Treg cells, with lymphoid tissues being higher than non-lymphoid tissues. This is consistent with the findings of *Tuovinen et al., 2006*, which indicate that the proportion of dual TCR Tregs in lymphoid tissues is higher than that of other types of T cells. This will contribute to a better understanding of the distribution characteristics of dual TCR Treg cells in different tissues and provide a basis for conducting functional experiments on dual TCR Treg cells at various tissue sites based on mRNA expression levels. And the proportion of liver and pancreas dual TCR Treg cells in non-lymphoid tissues is relatively high. In addition, our analysis of dual TCR T cell pairing types revealed significant differences between tissues. These findings not only suggest that lymphoid tissues have a high proportion of dual TCR Treg cells but also provide a comparative baseline and research direction for tissue-specific Treg cells in different locations.

The lower diversity of dual TCR Treg CDR3, the tissue preference in *V/J* gene usage, and the high clonal proliferation frequency in some tissues reflect the limited breadth of the dual TCR Treg subset response. It should be noted that the analysis of CDR3 diversity indicates that the TCR composition of dual TCR Treg cells is diverse, comparable to single TCR Treg cells. Due to differences in the number of single and dual TCR Treg cells in each sample, the diversity indices of single and dual TCR Treg in this study are not suitable for statistical comparative analysis. These cells may play a complementary role in specific physiological or pathological negative regulation, and the tissue-specific preference for *V/J* usage suggests that dual TCR Treg cells may have a characteristic lineage similar to that of γδ T cells. Current research indicates that γδ T cells in different tissues show differences in VJ subfamily usage due to directed migration after specific VJ recombination at different times (*Papadopoulou et al., 2020*). Dual TCR Treg cells may have certain lineage characteristics in V(D)J recombination and self-tolerance selection and migrate to settle in different tissues. The higher clonal proliferation

frequency of dual TCR Treg cells in the liver may exert a stronger inhibitory response to certain specific antigenic epitopes. The relatively higher immune tolerance of liver transplantation compared to other organ transplants may be related to the high proportion and clonal proliferation of dual TCR Treg cells in the liver. Based on the diversity and complexity of Treg functions, conducting a comparative analysis of the origins of dual TCR Treg cells and non-T cells with dual TCR will be a highly meaningful direction.

The majority of single/dual TCR Treg cells exhibit consistent expression of characteristic genes, and the high expression of certain genes in dual TCR Treg cells (*Foxp3*, *Foxo1*, and *Ctla4*) indicates that dual TCR Treg cells can function similarly to or more potently than single TCR Treg cells. In addition, specific or shared *Trbv* genes and mRNA molecule expression in dual TCR Treg cells has been identified in lymphoid or non-lymphoid tissues. Recently, Malte et al. used scATAC-seq technology to reveal organ-specific adaptation and conservation of tissue-resident immune cells such as Treg and Th17, and identify key transcription factors for multiple tissue-resident immune cells (*Simon et al., 2024*). Overall, the homogeneity and heterogeneity of these characteristic molecules within Treg cells suggest their potential induction in response to different antigenic epitopes and corresponding negative regulatory functions.

This study, based on single-cell RNA sequencing and T cell receptor (scRNA+TCR-seq) data of Treg cells in lymphatic and non-lymphatic tissues of mice under physiological conditions, reveals for the first time the phenomenon of a large number of dual TCR Treg cells in mice. Moreover, the CDR3 regions of these dual TCR Treg cells exhibit certain tissue- or lineage-specific characteristics in terms of VJ usage, diversity, and clonality. Compared to single TCR Treg cells, dual TCR Treg cells highly express immune-inhibitory related molecules, such as *Foxo3*, *Foxo1*, *Cd27*, *Il2ra*, *and Ikzf2*. Dual TCR Treg cells in LPL and skin also have more mRNA differentially expressed characteristic molecules. These results provide a new Treg cell subset (dual TCR Treg) for studying phenotypic characteristics, TCR composition characteristics, etc. This study delves into the complex biological effects and mechanisms of Treg cells from the perspective of TCR mRNA expression, providing a new perspective for Treg cell research and also offering technical solutions for decoding the TCR pairing and characteristics of dual TCR T cells. However, this study still has some shortcomings, such as the lack of protein data for individual dual TCR Treg cells and research on human dual TCR Treg cells. Subsequent studies can use dual receptor transgenic reporter mice (*Yang et al., 2020*), combined with FCM, tetramers, and scCITE-seq technology, to provide a foundation for research at the TCR protein expression level. We hope that more laboratories will participate in the study of dual TCR Treg cells, especially to further clarify the diversity and specificity of the immune negative regulatory role of Treg cells in human lymphatic and non-lymphatic tissues. Further experimental studies on their developmental differentiation, origins, functional gene expression, and effects will provide theoretical and technical means for the potential application of tissue-resident dual TCR Tregs.

## Materials and methods

### Research subjects and study samples

(1) Blood, kidney, liver, LPL, and pancreas tissues were collected from four C57BL/6 mice and CD4+Foxp3+ Thy1.1+ Treg cells were sorted for scRNA+TCR-seq analysis. (Source document: Oliver T et al. The tissue-resident regulatory T cell pool is shaped by transient multi-tissue migration and a conserved residency program. Immunity. 2024 Jul 9;57(7):1586–1602.e10. Shared Data Link: https://www.ncbi.nlm.nih.gov/geo/query/acc.cgi?acc=GSE266111.) (2) Spleen, iLN, and mLN tissues from four mice were collected and sorted into CD4+CD25+Treg cells, while skin tissues from four mice were collected and sorted into CD3+ T cells (both Treg and non-Treg). These four tissue-sorted cells were analyzed by scRNA+TCR-seq. (Source document: Nedwed et al. Using combined single-cell gene expression, TCR sequencing, and cell surface protein barcoding to characterize and track CD4+ T cell clones from murine tissues. Front Immunol. 2023 Oct 12;14:1241283. Shared Data Link: https://www.ncbi.nlm.nih.gov/geo/query/acc.cgi?acc=GSE240041.) (All mouse sequencing data comes from public databases, so ethical review is not involved.)

## The process of sharing raw data analysis

(1) Excel software was utilized for paired analysis of TCR α chain (A) and TCR β chain (B) from single Treg cell scTCR-seq results, including CD3 T cells in skin tissue. (2) In a single T cell, two types of chains 'can pair and assemble into single TCR' T cells named 'A+B'; three or more types of chains 'can pair and assemble into dual TCR' T cells named 'A+B1+B2; B+A1+A2; A1+A2+B1+B2'. (3) The proportions of total single and dual TCR Treg were calculated for each mouse or each tissue.

## Examples of single T-cell TCR pairing types

From the two datasets, single T cell was selected to display four different TCR pairing types, VDJ gene family names, and CDR3 AA sequences: A+B; A+B1+B2; B+A1+A2; and A1+A2+B1+B2.

## Analysis of TCR pairing types and V–J usage

Statistically analyze the proportions of five TCR T cell types in each sample, and use SPSS software to perform statistical analysis on the proportions of the three dual TCR T cell types and the single/dual TCR T cell ratios according to tissue origin; classify and calculate the proportions of all corresponding sequences in the 'β-V' and 'β-J' columns of the single and dual TCR T cell tables, and use SPSS software to conduct differential VJ usage analysis between single/dual TCR T cells. It should be noted that the dual and single TCR Treg cells analyzed in our results refer to T cells with functional alpha and beta chains that are paired. Treg cells that possess only a functional alpha or beta chains without TCR pairing, or those with non-functional alpha or beta chains participating in TCR pairing, were not included in our analysis.

## CDR3 diversity and clonality analysis

Use the inverse Simpson's index to assess TCR Treg cell diversity and perform statistical analysis; define clonal expansion as clone count ≥2, and analyze the proportion of clonal expansion of single/dual TCR Treg cells in the two datasets.

## CDR3 overlap analysis

The TRA CDR3 AA and TRB CDR3 AA of single and dual TCR Treg cells from different tissue sources were compared for overlap using the immunarch R package (version 0.9.0, https://github.com/immunomind/immunarch; *Balashov et al., 2024*) and the Jaccard index was used to analyze the frequency of overlap of CDR3 AA between each tissues pair; the first 10 high-frequency overlapping TRB CDR3 AA and TRA CDR3 AA sequences were traced and analyzed between single and dual TCR Treg cells from different tissue sources.

## Combined scRNA+TCR-seq analysis

The Seurat R package (version 5.0.1, https://satijalab.org/seurat/) was used for quality control of scRNA-seq results and preliminary removal of doublets. Data were excluded if nFeature >5000, nCount <20,000 or >1000, and percent.MT >5% (skin, spleen, iLN, and mLN tissue datasets) or 10% (blood, kidney, liver, LPL, and pancreas tissues). The DoubletFinder R package (version 2.0.4, https://github.com/chris-mcginnis-ucsf/DoubletFinder; *McGinnis and Liu, 2025*) was further utilized to accurately identify and remove doublets through steps such as simulating doublet generation, data dimensionality reduction, calculating the artificial nearest neighbor ratio (pANN), and setting thresholds. This process enhances the accuracy and reliability of data analysis. Furthermore, in the dataset containing five tissues with LPL, HTO labeling was incorporated (HTO labeling differentiates cells from different tissue sources by assigning unique barcodes to each cell; through the analysis of HTO labeling, double-positive and double-negative cells can be identified) to further exclude doublets and negative cells. T cells were then clustered and analyzed from different tissue sources after quality control. Based on the identical barcode identifiers of each T cell, the AddMetaData function was used to integrate scTCR-seq and scRNA-seq results, and only Treg cells meeting both criteria were included. Cells containing scTCR-seq and scRNA-seq data were displayed in UMAP plots and analyzed for the proportion of single and dual TCR Treg cells from different tissue sources. In this analysis, skin tissue was divided into Treg and non-Treg for comparison.

## Transcription factor expression analysis

Comparative analysis of the homogeneity of Treg cell signature molecules expressed in single and dual TCR Treg; differential expression analysis of the top 10 mRNA molecules in single tissue, overall single TCR Treg cells, and dual TCR Treg cells was conducted using the DESeq2 R package (version 1.38.3, https://github.com/thelovelab/DESeq2; *Love, 2025*) Comparative analysis of the heterogeneity of the expression of single and dual TCR Treg top 10 mRNA molecules in each tissue; comparative analysis of the heterogeneity of the expression of total single TCR Treg and total dual TCR Treg top 10 mRNA molecules.

## Statistical analysis

In this study, statistical analyses were conducted using R language and IBM SPSS Statistics 18 software. Independent sample *t*-tests were utilized to compare continuous variables between two groups, while chi-square tests were employed for categorical variables, with a p-value <0.05 considered statistically significant. For the analysis of single-cell RNA sequencing data, the DESeq2 R package was used to perform statistical analysis of differentially expressed genes, and the Wald test was applied to calculate the significance of gene expression differences. A $p_{adj} < 0.05$ was considered statistically significant.

## Acknowledgements

We would like to express our gratitude to *GEO* and *IMGT* databases for providing the availability of the data. Additionally, we thank Oliver T et al. and Nedwed et al. for carrying out the innovative study of Treg in mice and for sharing all original data. This study was supported by the National Natural Science Foundation of China (82471630&82160279) and the Guizhou Provincial Hundred-level Talent Fund [No.(2018)5637].

## Additional information

### Funding

| Funder | Grant reference number | Author |
|---|---|---|
| National Natural Science Foundation of China | 82471630 | Xinsheng Yao |
| National Natural Science Foundation of China | 82160279 | Xinsheng Yao |
| Guizhou Provincial Science and Technology Department | No. (2018)5637 | Xinsheng Yao |

The funders had no role in study design, data collection, and interpretation, or the decision to submit the work for publication.

### Author contributions

Yuanyuan Xu, Conceptualization, Formal analysis, Investigation, Visualization, Methodology, Writing - original draft, Writing - review and editing; Qi Peng, Long Ma, Jun Li, Conceptualization, Visualization, Methodology, Writing - original draft; Xiaoping Lu, Conceptualization, Visualization, Methodology; Xinsheng Yao, Conceptualization, Data curation, Supervision, Funding acquisition, Visualization, Methodology, Writing - original draft, Writing - review and editing

### Author ORCIDs

Yuanyuan Xu ⓘ http://orcid.org/0009-0008-5750-445X
Xinsheng Yao ⓘ https://orcid.org/0000-0001-6960-4003

Reviewer #2 (Public review): https://doi.org/10.7554/eLife.105504.4.sa1
Reviewer #3 (Public review): https://doi.org/10.7554/eLife.105504.4.sa2

Author response https://doi.org/10.7554/eLife.105504.4.sa3

# Additional files

### Supplementary files
MDAR checklist

### Data availability
All data analyzed in this study are deposited in the GEO database.

The following previously published datasets were used:

| Author(s) | Year | Dataset title | Dataset URL | Database and Identifier |
|---|---|---|---|---|
| Burton OT, Bricard O, Tareen S, Gergelits V, Andrews S, Biggins L, Roca CP, Whyte C, Junius S, Brajic A, Pasciuto E, Ali M, Lemaitre P, Schlenner SM, Ishigame H, Brown BD, Dooley J, Liston A | 2024 | The tissue-resident regulatory T cell pool is shaped by transient multi-tissue migration and a conserved residency program. scRNA-Seq profiling of mouse tissue Tregs with TCR sequencing and CITE-Seq | https://www.ncbi.nlm.nih.gov/geo/query/acc.cgi?acc=GSE266111 | NCBI Gene Expression Omnibus, GSE266111 |
| Nedwed AS, Helbich SS, Braband KL, Volkmar M, Delacher M, Marini F | 2023 | Using combined single-cell gene expression, TCR sequencing and cell surface protein barcoding to characterize and track CD4 T cell clones from murine tissues | https://www.ncbi.nlm.nih.gov/geo/query/acc.cgi?acc=GSE240041 | NCBI Gene Expression Omnibus, GSE240041 |

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
