## [Editor Report · eLife Assessment]

This study reanalyzed previously published scRNA-seq and TCR-seq data to examine the proportion and characteristics of dual-TCR-expressing Treg cells in mice, presenting some **useful** insights into TCR diversity and immune regulation. However, the evidence is **incomplete**, particularly with respect to data interpretation, statistical rigor, and the functionality of dual -TCR Treg cells. The study is potentially of interest to immunologists studying T-cell biology.

---

## [Referee Report · Reviewer #2 (Public review)]

Summary:

The manuscript, by Xu and Peng, et al. investigates whether co-expression of 2 T cell receptor (TCR) clonotypes can be detected in FoxP3+ regulatory CD4+ T cells (Tregs) and if it is associated with identifiable phenotypic effects. This paper presents data reanalyzing publicly available single-cell TCR sequencing and transcriptional analysis, convincingly demonstrating that dual TCR co-expression can be detected in Tregs, both in peripheral circulation as well as among Tregs in tissues. They then compare metrics of TCR diversity between single-TCR and dual TCR Tregs, as well as between Tregs in different anatomic compartments, finding the TCR repertoires to be generally similar though with dual TCR Tregs exhibiting a less diverse repertoire and some moderate differences in clonal expansion in different anatomic compartments. Finally, they examine the transcriptional profile of dual TCR Tregs in these datasets, finding some potential differences in expression of key Treg genes such as Foxp3, CTLA4, Foxo3, Foxo1, CD27, IL2RA, and Ikzf2 associated with dual TCR-expressing Tregs, which the authors postulate implies a potential functional benefit for dual TCR expression in Tregs.

Strengths:

This report examines an interesting and potentially biologically significant question, given recent demonstrations that dual TCR co-expression is a much more common phenomenon than previously appreciated (approximately 15-20% of T cells) and that dual TCR co-expression has been associated with significant effects on the thymic development and antigenic reactivity of T cells. This investigation leverages large existing datasets of single-cell TCRseq/RNAseq to address dual TCR expression in Tregs. The identification and characterization of dual TCR Tregs is rigorously demonstrated and presented, providing convincing new evidence of their existence.

Weaknesses:

The existence of dual TCR expression by Tregs has previously been demonstrated in mice and humans, limiting the novelty of the reported findings. The presented results should be considered in the context of these prior important findings. The focus on self-citation of their previous work, using the same approach to measure dual TCR expression in other datasets. limits the discussion of other more relevant and impactful published research in this area. Also, Reference #7 continues to list incorrect authors. The authors do not present a balanced or representative description of the available knowledge about either dual TCR expression by T cells or TCR repertoires of Tregs.

The approach used follows a template used previously by this group for re-analysis of existing datasets generated by other research groups. The descriptions and interpretations of the data as presented are still shallow, lacking innovative or thoughtful approaches that would potentially be innovation or provide new insight.

This demonstration of dual TCR Tregs is notable, though the authors do not compare the frequency of dual TCR co-expression by Tregs with non-Tregs. This limits interpreting the findings in the context of what is known about dual TCR co-expression in T cells. The response to this criticism in a previous review is considered non-responsive and does not improve the data or findings.

Comparison of gene expression by single- and dual TCR Tregs is of interest, but as presented is difficult to interpret. The interpretations of the gene expression analyses are somewhat simplistic, focusing on single-gene expression of some genes known to have function in Tregs. However, the investigators continue to miss an opportunity to examine larger patterns of coordinated gene expression associated with developmental pathways and differential function in Tregs (Yang. 2015. Science. 348:589; Li. 2016. Nat Rev Immunol. Wyss. 2016. 16:220; Nat Immunol. 17:1093; Zenmour. 2018. Nat Immunol. 19:291). No attempt to define clusters is made. No comparison is made of the proportions of dual TCR cells in transcriptionally-defined clusters. The broad assessment of key genes by single- and dual TCR cells is conceptually interesting, but likely to be confounded by the heterogeneity of the Treg populations. This would need to be addressed and considered to make any analyses meaningful.

The study design, re-analysis of existing datasets generated by other scientific groups, precludes confirmation of any findings by orthogonal analyses.

---

## [Referee Report · Reviewer #3 (Public review)]

Summary:

This study addressed the TCR pairing types and CDR3 characteristics of Treg cells. By analyzing scRNA and TCR-seq data, it claims that 10-20% of dual TCR Treg cells exist in mouse lymphoid and non-lymphoid tissues and suggests that dual TCR Treg cells in different tissues may play complex biological functions.

Strengths:

The study addresses an interesting question of how dual-TCR-expressing Treg cells play roles in tissues.

Weaknesses:

This study is inadequate, particularly regarding data interpretation, statistical rigor, and the discussion of the functional significance of Dual TCR Tregs.

Comments on revisions:

Although the authors have provided brief explanations in response to the reviewers' comments, they do not present any additional analyses that would address the fundamental concerns in a convincing manner.

Moreover, the in silico analyses presented in the manuscript alone are insufficient to support the conclusions, and the functional experiments requested by the reviewers have not been conducted.

In the current rebuttal, while some textual additions have been made to the manuscript, the only substantial revision to the figures appears to be the inclusion of statistical significance annotations (e.g., Fig. 1G, Fig. 3G). These changes do not adequately strengthen the overall data or address the core issues raised.

---

## [Author Response]

The following is the authors’ response to the previous reviews

**Reviewer #1 (Public review):**
(1) The use of single-cell RNA and TCR sequencing is appropriate for addressing potential relationships between gene expression and dual TCR.

Thank you for your detailed review and suggestions. The main advantages of scRNA+TCR-seq are as follows: (1) It enables comparative analysis of features such as the ratio of single TCR paired T cells to dual TCR paired T cells at the level of a large number of individual T cells, through mRNA expression of the α and β chains. In the past, this analysis was limited to a small number of T cells, requiring isolation of single T cells, PCR amplification of the α and β chains, and Sanger sequencing; (2) While analyzing TCR paired T cell characteristics, it also allows examination of mRNA expression levels of transcription factors in corresponding T cells through scRNA-seq.

(2) The data confirm the presence of dual TCR Tregs in various tissues, with proportions ranging from 10.1% to 21.4%, aligning with earlier observations in αβ T cells.

Thank you very much for your detailed review and suggestions. Early studies on dual TCR αβ T cells have been very limited in number, with reported proportions of dual TCR T cells ranging widely from 0.1% to over 30%. In contrast, scRNA+TCR-seq can monitor over 5,000 single and paired TCRs, including dual paired TCRs, in each sample, enabling more precise examination of the overall proportion of dual TCR αβ T cells. It is important to note that our analysis focuses on T cells paired with functional α and β chains, while T cells with non-functional chain pairings and those with a single functional chain without pairing were excluded from the total cell proportion analysis. Previous studies generally lacked the ability to determine expression levels of specific chains in T cells without dual TCR pairings.

(3) Tissue-specific patterns of TCR gene usage are reported, which could be of interest to researchers studying T cell adaptation, although these were more rigorously analyzed in the original works.

Thank you very much for your detailed review and suggestions. T cell subpopulations exhibit tissue specificity; thus, we conducted a thorough investigation into Treg cells from different tissue sites. This study builds upon the original by innovatively analyzing the differences in VDJ rearrangement and CDR3 characteristics of dual TCR Treg cells across various tissues. This provides new insights and directions for the potential existence of “new Treg cell subpopulations” in different tissue locations. The results of this analysis suggest the necessity of conducting functional experiments on dual TCR Treg cells at both the TCR protein level and the level of effector functional molecules.

(4) Lack of Novelty: The primary findings do not substantially advance our understanding of dual TCR expression, as similar results have been reported previously in other contexts.

Thank you for your detailed review and suggestions. Early research on dual TCR T cells primarily relied on transgenic mouse models and in vitro experiments, using limited TCR alpha chain or TCR beta chain antibody pairings. Flow cytometry was used to analyze a small number of T cells to estimate dual TCR T cell proportion. No studies have yet analyzed dual TCR Treg cell proportion, V(D)J recombination, and CDR3 characteristics at high throughput in physiological conditions. The scRNA+TCR-seq approach offers an opportunity to conduct extensive studies from an mRNA perspective. With high-throughput advantages of single-cell sequencing technology, researchers can analyze transcriptomic and TCR sequence characteristics of all dual TCR Treg cells within a study sample, providing new ideas and technical means for investigating dual TCR T cell proportions, characteristics, and origins under different physiological and pathological states.

(5) Incomplete Evidence: The claims about tissue-specific differences lack sufficient controls (e.g., comparison with conventional T cells) and functional validation (e.g., cell surface expression of dual TCRs).

Thank you for your detailed review and suggestions. This study indeed only analyzed dual TCR Treg cells from different tissue locations based on the original manuscript, without a comparative analysis of other dual TCR T cell subsets corresponding to these tissue locations. The main reason for this is that, in current scRNA+TCR-seq studies of different tissue locations, unless specific T cell subsets are sorted and enriched, the number of T cells obtained from each subset is very low, making a detailed comparative analysis impossible. In the results of the original manuscript, we observed a relatively high proportion of dual TCR Treg cell populations in various tissues, with differences in TCR composition and transcription factor expression. Following the suggestions, we have included additional descriptions in R1, citing the study by Tuovinen et al., which indicates that the proportion of dual TCR Tregs in lymphoid tissues is higher than other T cell types. This will help understand the distribution characteristics of dual TCR Treg cells in different tissues and provide a basis for mRNA expression levels to conduct functional experiments on dual TCR Treg cells in different tissue locations.

(6) Methodological Weaknesses: The diversity analysis does not account for sample size differences, and the clonal analysis conflates counts and clonotypes, leading to potential misinterpretation.

We thank you for your review and suggestions. In response to your question about whether the diversity analysis considered the sample size issue, we conducted a detailed review and analysis. This study utilized the inverse Simpson index to evaluate TCR diversity of Treg cells. A preliminary analysis compared the richness and evenness of single TCR Treg cell and dual TCR Treg cell repertoires. The two datasets analyzed were from four mouse samples with consistent processing and sequencing conditions. However, when analyzing single TCR Tregs and dual TCR Tregs from various tissues, differences in detected T cell numbers by sequencing cannot be excluded from the diversity analysis. Following recommendations, we provided additional explanations in R1: CDR3 diversity analysis indicates TCR composition of dual TCR Treg cells exhibits diversity, similar to single TCR Treg cells; however, diversity indices of single TCR Tregs and dual TCR Tregs are not suitable for statistical comparison. Regarding the "clonal analysis" you mentioned, we define clonality based on unique TCR sequences; cells with identical TCR sequences are part of the same clone, with ≥2 counts defined as expansion. For example, in Blood, there are 958 clonal types and 1,228 cells, of which 449 are expansion cells. In R1, we systematically verified and revised clonal expansion cells across all tissue samples according to a unified standard.

(7) Insufficient Transparency: The sequence analysis pipeline is inadequately described, and the study lacks reproducibility features such as shared code and data.

Thank you for your review and suggestions. Based on the original manuscript, we have made corresponding detailed additions in R1, providing further elaboration on the analysis process of shared data, screening methods, research codes, and tools. This aims to offer readers a comprehensive understanding of the analytical procedures and results.

(8) Weak Gene Expression Analysis: No statistical validation is provided for differential gene expression, and the UMAP plots fail to reveal meaningful clustering patterns.

Thank you very much for your review and suggestions. Based on your recommendations, we conducted an initial differential expression analysis of the top 10 mRNA molecules in single TCR Treg and dual TCR Treg cells using the DESeq2 R package in R1, with statistical significance determined by Padj < 0.05. Regarding the clustering patterns in the UMAP plots, since the analyzed samples consisted of isolated Treg cell subpopulations that highly express immune suppression-related genes, we did not perform a more detailed analysis of subtypes and expression gene differences. This study primarily aims to explore the proportions of single TCR and dual TCR Treg cells from different tissue sources, as well as the characteristics of CDR3 composition, with a focus on showcasing the clustering patterns of samples from different tissue origins and various TCR pairing types.

(9) A quick online search reveals that the same authors have repeated their approach of reanalysing other scientists' publicly available scRNA-VDJ-seq data in six other publications,In other words, the approach used here seems to be focused on quick re-analyses of publicly available data without further validation and/or exploration.

Thank you for your review and suggestions. Most current studies utilizing scRNA+TCR-seq overlook analysis of TCR pairing types and related research on single TCR and dual TCR T cell characteristics. Through in-depth analysis of shared scRNA+TCR-seq data from multiple laboratories, we discovered a significant presence of dual TCR T cells in high-throughput T cell research results that cannot be ignored. In this study, we highlight the higher proportion of dual TCR Tregs in different tissue locations, which exhibits a certain degree of tissue specificity, suggesting these cells may participate in complex functional regulation of Tregs. This finding provides new ideas and a foundation for further research into dual TCR Treg functions. However, as reviewers pointed out, findings from scRNA+TCR-seq at the mRNA level require additional functional experiments on dual TCR T cells at the protein level. We have supplemented our discussion in R1 based on these suggestions.

**Reviewer #2 (Public review):**
(1) The existence of dual TCR expression by Tregs has previously been demonstrated in mice and humans (Reference #18 and Tuovinen. 2006. Blood. 108:4063; Schuldt. 2017. J Immunol. 199:33, both omitted from references). The presented results should be considered in the context of these prior important findings.

Thank you very much for your review and suggestions. Based on the original manuscript, we have supplemented our reading, understanding, and citation of closely related literature (Tuovinen, 2006, Blood, 108:4063 (line 44,line175 in R1); Schuldt, 2017, J Immunol, 199:33 (line 44,line178 in R1)). We once again appreciate the valuable comments from the reviewers, and we will refer to these in our subsequent dual TCR T cell research.

(2) This demonstration of dual TCR Tregs is notable, though the authors do not compare the frequency of dual TCR co-expression by Tregs with non-Tregs. This limits interpreting the findings in the context of what is known about dual TCR co-expression in T cells.

Thank you very much for your review and suggestions. This analysis is primarily based on the scRNA+TCR-seq study of sorted Treg cells, where we found the proportions and distinguishing features of dual TCR Treg cells in different tissue sites. Given the diversity and complexity of Treg function, conducting a comparative analysis of the origins of dual TCR Treg cells and non-T cells with dual TCRs will be a meaningful direction. Currently, peripheral induced Treg cells can originate from the conversion of non-Treg cells; however, little is known about the sources and functions of dual TCR Treg cell subsets in both central and peripheral sites. In R1, we have supplemented the discussion regarding the possible origins and potential applications of the "novel dual TCR Treg" subsets.

(3) Comparison of gene expression by single- and dual TCR Tregs is of interest, but as presented is difficult to interpret. Statistical analyses need to be performed to provide statistical confidence that the observed differences are true.

Thank you very much for your review and suggestions. Based on your recommendations, we performed an initial differential expression analysis of the top 10 mRNA molecules in single TCR Treg and dual TCR Treg cells using the DESeq2 R package in R1, with a statistical significance threshold of Padj<0.05 for comparisons.

(4) The interpretations of the gene expression analyses are somewhat simplistic, focusing on the single-gene expression of some genes known to have a function in Tregs. However, the investigators miss an opportunity to examine larger patterns of coordinated gene expression associated with developmental pathways and differential function in Tregs (Yang. 2015. Science. 348:589; Li. 2016. Nat Rev Immunol. Wyss. 2016. 16:220; Nat Immunol. 17:1093; Zenmour. 2018. Nat Immunol. 19:291).

Thank you for your review and suggestions. This study is based on publicly available scRNA+TCR-seq data from different organ sites generated by the original authors, focusing on sorted and enriched Treg cells within each tissue sample. However, there was no corresponding research on other cell types in each tissue sample, preventing analysis of other cells and factors involved in development and differentiation of single TCR Treg and dual TCR Treg. The literature suggested by the reviewer indicates that development, differentiation, and function of Treg cells have been extensively studied, resulting in significant advances. It also highlights complexity and diversity of Treg origins and functions. This research aims to investigate "novel dual TCR Treg cell subpopulations" that may exhibit tissuespecific differences found in the original authors' studies of Treg cells across different organ sites. This suggests further experimental research into their development, differentiation, origin, and functional gene expression as an important direction, which we have supplemented in the discussion section of R1.

**Reviewer #3 (Public review):**
(1) Definition of Dual TCR and Validity of Doublet Removal:This study analyzes Treg cells with Dual TCR, but it is not clearly stated how the possibility of doublet cells was eliminated. The authors mention using DoubletFinder for detecting doublets in scRNA-seq data, but is this method alone sufficient?We strongly recommend reporting the details of doublet removal and data quality assessment in the Supplementary Data.

Thank you very much for your review and suggestions. In the analysis of the shared scRNA+TCR-seq data across multiple laboratories, as you mentioned, this study employed the DoubletFinder R package to exclude suspected doublets. Additionally, we used the nCount values of individual cells (i.e., the total sequencing reads or UMI counts for each cell) as auxiliary parameters to further optimize the assessment of cell quality. Generally, due to the possibility that doublet cells may contain gene expression information from two or more cells, their nCount values are often abnormally high. In this study, all cells included in the analysis had nCount values not exceeding 20,000. Among the five tissue sample datasets, we further utilized hashtag oligonucleotide (HTO) labeling (where HTO labeling provides each cell with a unique barcode to differentiate cells from different tissue sources. By analyzing HTO labels, doublets and negative cells can be accurately identified) to eliminate doublets and negative cells.After the removal of chimeric cells, all samples exhibited T cells that possessed two or more TCR clones. This phenomenon validates the reliability of the methodological approach employed in this study and indicates that the analytical results accurately reflect the proportion of dual TCR T cells. Based on the recommendations of the reviewers, we have supplemented and clarified the methods and discussion sections in the manuscript. It is particularly noteworthy that in our analysis, the discussed dual TCR Treg cells and single TCR Treg cells specifically refer to those T cells that possess both functional α and β chains, which are capable of forming TCR. We have excluded from this analysis any Treg cells that possess only a single functional α or β chain and do not form TCR pairs, as well as those Treg cells in which the α or β chains involved in TCR pairing are non-functional.

(2) In Figure 3D, the proportion of Dual TCR T cells (A1+A2+B1+B2) in the skin is reported to be very high compared to other tissues. However, in Figure 4C, the proportion appears lower than in other tissues, which may be due to contamination by non-Tregs. The authors should clarify why it was necessary to include non-Tregs as a target for analysis in this study. Additionally, the sensitivity of scRNA-seq and TCR-seq may vary between tissues and may also be affected by RNA quality and sequencing depth in skin samples, so the impact of measurement bias should be assessed.

We deeply appreciate your review and constructive comments. Based on the original manuscript, we have further supplemented and elaborated on the uniqueness and relative proportions of double TCR T cell pairs in skin tissue samples in Section R1. Due to the scarcity of T cells in skin samples, we included some non-Treg cells during single-cell RNA sequencing and TCR sequencing to obtain a sufficient number of cells for effective analysis. The presence of non-regulatory T cells may indeed impact the statistical representation of double TCR T cells as well as the related comparative analyses, as noted by the reviewer. T cells with A1+A2+B1+B2 type double TCR pairings are primarily found within the non-regulatory T cell population in the skin. In response to this point, we have provided a detailed explanation of this analytical result in the revised manuscript R1. Furthermore, concerning the two datasets included in the study, we conducted a comparative analysis in R1, exploring how factors such as sequencing depth at different tissue sites might introduce biases in our findings, which we have thoroughly elaborated upon in the discussion section. We thank you once again for your valuable suggestions.

(3) Issue of Cell Contamination:In Figure 2A, the data suggest a high overlap between blood, kidney, and liver samples, likely due to contamination. Can the authors effectively remove this effect? If the dataset allows, distinguishing between blood-derived and tissue-resident Tregs would significantly enhance the reliability of the findings. Otherwise, it would be difficult to separate biological signals from contamination noise, making interpretation challenging.

We thank you for your review and suggestions. We have carefully verified data sources for tissues such as blood, kidneys, and liver. In the study by Oliver T et al., various techniques were employed to differentiate between leukocytes from blood and those from tissues, ensuring accurate identification of leukocytes from tissue samples. First, anti-CD45 antibody was injected intravenously to label cells in the vasculature, verifying that analyzed cells were indeed resident in the tissue. Second, prior to dissection and cell collection, authors performed perfusion on anesthetized mice to reduce contamination of tissue samples by leukocytes from the vasculature. Additionally, during single-cell sequencing, authors utilized HTO technology to avoid overlap between cells from different tissues.

Analysis of the scRNA+TCR-seq data shared by the original authors revealed highly overlapping TCR sequences in blood, kidney, and liver, despite distinct cell labels associated with each tissue. While these techniques minimize overlap of cells from different sources, they cannot completely rule out the potential impact of this technical issue. As suggested, we have provided additional clarification in R1 of the manuscript regarding this phenomenon of high overlap in the kidney, liver, and blood, indicating that the possibility of Treg migration from blood to kidney and liver cannot be entirely excluded.

(4) Inconsistency Between CDR3 Overlap and TCR Diversity:The manuscript states that Single TCR Tregs have a higher CDR3 overlap, but this contradicts the reported data that Dual TCR Tregs exhibit lower TCR diversity (higher 1/DS score). Typically, when TCR diversity is low (i.e., specific clones are concentrated), CDR3 overlap is expected to increase. The authors should carefully address this discrepancy and discuss possible explanations.

Thank you for your review and suggestions. Regarding the potential relationship between CDR3 overlap and TCR diversity, in samples with consistent sequencing depth, lower diversity indeed corresponds to a higher proportion of CDR3 overlap. In our analysis of scRNA+TCR-seq data, we found that single TCR Tregs exhibit both higher diversity and CDR3 overlap, seemingly presenting contradictory analytical results (i.e., dual TCR Tregs show lower TCR diversity and CDR3 overlap). In R1, we supplemented the analysis of possible reasons: the presence of multiple TCR chains in dual TCR Treg cells may lead to a higher uniqueness of CDR3 due to multiple rearrangements and selections, resulting in lower CDR3 overlap; the lower diversity of dual TCR Tregs may be related to the number of T cells sequenced in each sample. The CDR3 diversity analysis in this study merely suggests that the TCR composition of dual TCR Treg cells is diverse, similar to that of single TCR Tregs. However, the diversity indices of single TCR Tregs and dual TCR Tregs are not suitable for statistical comparative analysis. A more in-depth and specific analysis of the diversity and overlap of the VDJ recombination mechanisms and CDR3 composition in dual TCR Tregs during development will be an important technical means to elucidate the function of dual TCR Treg cells.

(5) Functional Evaluation of Dual TCR Tregs:This study indicates gene expression differences among tissue-resident Dual TCR T cells, but there is no experimental validation of their functional significance. Including functional assays, such as suppression assays or cytokine secretion analysis, would greatly enhance the study's impact.

We sincerely appreciate your review and suggestions: In this analysis of scRNA+TCR-seq data, we innovatively discovered a higher proportion of dual TCR Treg cells in different tissue sites, which exhibited differences in tissue characteristics. Furthermore, we conducted a comparative analysis of the homogeneity and heterogeneity between single TCR Treg and dual TCR Treg cells. This result provides a foundation for further research on the origin and characteristics of dual TCR Treg cells in different tissue sites, offering new insights for understanding the complexity and functional diversity of Treg cells. Based on your suggestions, we have supplemented R1 with the feasibility of further exploring the functions of tissue-resident dual TCR T cells and the necessity for potential application research.

(6) Appropriateness of Statistical Analysis:When discussing increases or decreases in gene expression and cell proportions (e.g., Figure 2D), the statistical methods used (e.g., t-test, Wilcoxon, FDR correction) should be explicitly described. They should provide detailed information on the statistical tests applied to each analysis.

Thank you for your review and suggestions: Based on the original manuscript, we have supplemented the specific statistical methods for the differences in cell proportions and gene expression in R1.